# Demystifying Case Management in Aotearoa New Zealand: A Scoping and Mapping Review

**DOI:** 10.3390/ijerph20010784

**Published:** 2022-12-31

**Authors:** Caroline Stretton, Wei-Yen Chan, Dianne Wepa

**Affiliations:** 1Centre for Person Centred Research (PCR), School of Clinical Sciences, Faculty of Health and Environmental Sciences, AUT University, North Shore, Auckland 1142, New Zealand; 2School of Public Health and Interdisciplinary Studies, Faculty of Health and Environmental Sciences, AUT University, North Shore, Auckland 1142, New Zealand; 3School of Nursing & Healthcare Leadership, Faculty of Health Studies, University of Bradford, Bradford BD7 1DP, UK; 4Mental Health and Suicide Prevention Research and Education Group, Clinical and Health Sciences, University of South Australia, Adelaide, SA 5000, Australia

**Keywords:** case management, navigation, integrated care, coordinated care, case managers, social ecology maps, interprofessional practice, whānau ora, New Zealand, complexity

## Abstract

Background: Community-based case managers in health have been compared to glue which holds the dynamic needs of clients to a disjointed range of health and social services. However, case manager roles are difficult to understand due to poorly defined roles, confusing terminology, and low visibility in New Zealand. Aim: This review aims to map the landscape of case management work to advance workforce planning by clarifying the jobs, roles, and relationships of case managers in Aotearoa New Zealand (NZ). Methods: Our scoping and mapping review includes peer-reviewed articles, grey literature sources, and interview data from 15 case managers. Data was charted iteratively until convergent patterns emerged and distinctive roles identified. Results: A rich and diverse body of literature describing and evaluating case management work in NZ (*n* = 148) is uncovered with at least 38 different job titles recorded. 18 distinctive roles are further analyzed with sufficient data to explore the research question. Social ecology maps highlight diverse interprofessional and intersectoral relationships. Conclusions: Significant innovation and adaptations are evident in this field, particularly in the last five years. Case managers also known as health navigators, play a pivotal but often undervalued role in NZ health care, through their interprofessional and intersectoral relationships. Their work is often unrecognised which impedes workforce development and the promotion of person-centered and integrated health care.

## 1. Introduction

Case managers within a health context have been likened to relational glue which joins the personal and fluctuating needs of clients to a disjointed range of health and social services [1]. The role of the case manager though, particularly in New Zealand (NZ), is poorly understood [2] and lacks visibility [3]. As a collaborative relational process, the work of case management involves a series of stages that include engagement, holistic assessment, planning, education, training and skills development, emotional and motivational support, advising, coordination, and monitoring [4]. International evidence supports the effectiveness of community-based case management approaches in certain situations to improve health outcomes [5] and professional standards of practice have been developed [6]. However, in NZ this role appears to have evolved differently from other countries like the United States, where most case managers are nurses with post-graduate qualifications [7]. In NZ, having a recognised health qualification or experience may not be a requirement for employment in some case management roles and fewer options for workforce training are available [7]. The reasons why case management is not well understood may be due to conceptual confusion, significant variation in terms used to describe the role, and a lack of clarity in the role. Terminological variance, role vagueness and role ambiguity are well recognised problems which plague the international case management literature [8,9,10]. It is likely to be similar in our country but there is minimal NZ workforce data is to confirm this [2,3,11].

This difficulty defining the key concepts of case management can create confusion on several fronts- (a glossary in Appendix B provides additional description of key concepts contained in this review). The phrase ‘case management’ is commonly used in three ways: (1) to describe the ‘work’ of case management (i.e., the underlying change process where client and/or their extended family are supported to attain goals [12]); (2) for the ‘job title’-the person who is doing the case management work (i.e., the case manager); or (3) for the ‘healthcare strategy’ of using a targeted and integrated approach to optimize the quality of health care delivery to particular clients with high needs [13]. As well, the job title of a case manager varies widely. In New Zealand these workers may be called navigators [14], local area coordinators [15], or care managers [16]; or in response to the development of indigenous culturally responsive models of case management, as kaiārahi [17], kaitautoko [18] or kaimanaaki [19]. Māori are the indigenous peoples of New Zealand. Box 1 contains an explanation of common job titles that use concepts from a Māori world view to describe these roles. Aside from imprecise phrasing and terminological variance in job titles, further additional factors can produce role confusion.
Box 1Commonly used Māori language (te reo) job titles for case management/health navigation roles.
**Term****Definition**Commonly usedKaiārahiOne who navigatesOften usedKaimanaaki Kaimanaaki whānauSupport person Support person to the client/patient and whānau/extended familyOccasionally usedKaiwhakatere Kaiwhakahaere Kaiwhakaara Paeārahi Kaiwhakamana Kaitūhono KaitautokoNavigator Coordinator Organiser Intermediary Support Person Connector Support Person

Lack of role clarity, confusion and ambiguity [20] may be also due to the varied contexts that case managers work in. In New Zealand, health-related case managers are employed by governmental organisations such as District Health Boards with the Ministry of Health and Accident Compensation Corporation (ACC) (a no-faults social insurance provider); non-governmental organisations like community based mental health providers, primary health care practices and Māori and Pacific organisations. Case managers can also work in the private insurance sector [7]. Their divergent legal and financial obligations drive different goals and behaviour including the need to respond to requirements related to codes of health and disability ethics, ACC legislation, primary health care contracts and professional legislation, mental health legislation, or contractual requirements of private insurance entitlements. Clients receiving case management are also likely to have complex health and social needs, and limited capacity to access health services [21]. These clients are likely to live with differing health conditions such as physical injury, mental health conditions, multiple long-term conditions like diabetes, and cancer, aging-related conditions and families and individuals who experience significant socio-economic disadvantage. Therefore, the purpose and function of the case management role may vary depending on the type of employer, client characteristics, and other related contextual factors.

A further source of variation is the type of case management practice model being implemented. The case management literature describes numerous models. The most common models include the brokerage, clinical, strengths, and intensive models [1,10]. The simplest model is the brokerage model which focuses on needs evaluation, referral to services, and oversight of care coordination. In this model the client is a passive recipient of health services and relies on the case manager to signpost services to them. The clinical model involves the case manager providing therapeutic input with greater awareness of individual health needs. Care is tailored to enable access to services. The strengths-based approach supports the client’s capability to work in partnership with the case manager to attain their stated goals and navigate through the health. With an intensive model, the case manager provides a high level of support for a defined period to meet a specific need. Systematically clarifying similarities and differences between different types of case management roles in NZ should reduce confusion and enhance clarity of different roles and potentially identify ways to promote more coherent and consistent terminology.

The principles and processes of case management have the potential to make a unique contribution towards the integration of health care, social services and other sector services by providing support for client-centered care for people with complex health conditions [22,23,24,25]. Addressing the problem of health system fragmentation and persisting inequities in NZis an urgent priority for the health sector [24]. A comprehensive report by the NZ Productivity Commission concluded that improved societal outcomes would be drawn from services that targeted the most disadvantaged clients who experienced high complex health needs and low capacity to navigate the system (see bottom right quadrant in Figure 1 below). These clients are likely to need to engage with a navigator to find optimal solutions for their complex issues [26]. The Commission’s report highlights the valued role of health navigators. Nevertheless, despite the availability of this report, heath navigators, care coordinators, and case managers are not visible within related reports such as health workforce documentation [2,3] or social service workforce planning in NZ [27].

A case management approach, if implemented effectively, can reduce the cost and use of tertiary services such as hospital stays and improve the experience of users and carers [25]. However, role vagueness, role confusion, and role ambiguity can act as barriers to effective implementation of case management approaches and undermine job satisfaction and professional development for case managers themselves [20]. Therefore, this paper aims to examine the different job titles to describe case management roles; explore how different types of case managers function and investigate the network of relationships that underpin interprofessional and intersectoral case management practices. The purpose of the review was to scope and map the landscape of case management work in New Zealand as a strategy to advance future workforce development. Our research question was: ‘what are the jobs, roles, and relationships of case managers in Aotearoa New Zealand?

## 2. Materials and Methods

Scoping reviews [28,29,30] are commonly used to map key concepts that influence a phenomenon of interest, identify chief sources of data exploring the topic and “examine the extent, range and nature” of evidence ([29], p. 12) particularly if the area being examined is complex or has not been systematically reviewed before. We used the recommended six stages of scoping reviews [28,29] and the PRSIM ScR guidelines to guide our review [30]. The purpose of this review was not to appraise the quality of evidence found or provide in-depth analysis. Instead, this review seeks to provide a comprehensive overview, to improve conceptual clarity, and assist practitioners, educators, and policy makers to make sense of this area within the NZ context.

### 2.1. Stage One: Identifying the Research Question

The PCC framework was used to identify the population (P), concepts (C) and context (C) of this study [31] and help develop the overall research question and inclusion criteria. The population of interest included people who were currently working as case managers or performing case management work in NZ. Drawing on the work of Lukersmith [32] we adapted a definition for the NZ context and in this review defined a case manager as a named person or team, who provides continuity of care, and plans and organizes the coordination of health care services. Key concepts of this review were: job titles, continuity of care [33], the purpose of each role [32], and the specific actions undertaken by case managers [4], and relationships that case managers had with clients, other health care workers and organisations across the health and social sector (See Table 1 and for more detail the glossary A). We looked for any sources which contained information describing any of these concepts for health-related case management services for clients living in the community. A role was considered ‘health-related’ if the service received any funding from the Ministry of Health. The study focused on community health settings. We chose this context to enable comparison with the review of Lukersmith [34], and because we anticipated case managers in the community would have more diverse social networks than those who have clients who are still inpatient hospital and residential settings.

### 2.2. Stage Two Indentifying Relevant Sources

Due to the conceptual ambiguity of the topic, our initial attempts (before starting this review), to search, find and review NZ-based literature on case management had uncovered limited data. To overcome this problem, we designed our scoping review to gather data from three different areas: health databases, grey literature, and case manager interviews. We extended our search beyond peer-reviewed health databases to include grey literature because we wanted a wide-ranging and more complete view of all available evidence [35]. We anticipated that despite a comprehensive search strategy, gaps in knowledge would remain. Therefore, we elected to also interview case managers to supplement our dataset by providing real-world and context specific insights into current case management practice issues [36].

We completed 15 interviews with case managers who were working as a case manager or employing community-based case managers. Participants were recruited through the personal and professional networks of the research team and the project advisory group. A diversity of perspectives was sought for this project. Participants were chosen to provide different perspectives on the following: service contexts, legal jurisdiction, theoretical frameworks, and client characteristics. Purposeful sampling was used to ensure that participants worked in a variety of roles. Interviews took around 60 min and took place at a time chosen by the participant which was often after hours outside of work time. Each participant received a supermarket voucher in appreciation of their time. Approval for the interviews was provided by AUTEC and an oral consent protocol used (AUTEC Reference number 21/439). Interviews were conducted online using video conferencing and were video recorded. Interviews were semi-structured, and the interview guide was based on the analytic concepts described in Table 1 and adapted from the work of Lukersmith [32]. During the interview participants answered question about their role, what they understood the purpose and function, key client characteristics, any theoretical models they used and the actions that they undertook in their role. Participants were also asked to reflect on the case management taxonomy in relation to their specific work context [4]. Table 2 contains brief information about the type of organisation participants worked for and the type of case management work that they did.

### 2.3. Stage Three: Source Selection

#### 2.3.1. Search Strategies

The first search focused on health databases. Peer-reviewed literature was sourced systematically across several databases: CINAHL (via EBSCO), Medline (via EBSCO), ERIC (via Ovid), Scopus, and Google Scholar. Studies were included if they contained data that referred to someone working as a case manager or doing any case management work in a community healthcare setting in NZ. Literature of any type could be included if it contained some information related to jobs, roles (functions and purpose) or relationships of case managers. We limited our search to articles after the year 2000 to allow a wide range of data to be included but still ensure relevance. Studies needed to be in English. Health databases were searched between November 2021 and March 2022. Following the search, one author (WYC) imported citations into Covidence software [37]. Covidence is a web-based collaboration software platform that streamlines the production of systematic and other literature reviews. Two independent reviewers screened citations against the inclusion criteria (WYC, DW). Full-text appraised articles were selected if they met the inclusion criteria and if there was a disagreement then the team met to discuss whether the article should be included. Appendix C describes the search strategy used for the peer reviewed health database searching.

The second search focused on grey literature. Grey literature refers to literature “produced at all levels of government, academics, business, industry in print and electronic formats but which is not controlled by commercial publishers” [38]. Specifically for this study it included reports and evaluations from governmental and non-governmental agencies, job advertisements or descriptions, information from health care providers on the open internet, videos and presentations. A structured search plan for grey literature was developed in conjunction with information specialists [35,39]. Grey literature was searched using the open internet and included the Ministry of Health, Citizens Advice Bureau, NZ Research, and targeted job websites including Seek, Indeed, TradeMe Jobs and LinkedIn. The grey literature was searched between November 2021 and March 2022. details of the search strategy are contained in Appendix C.

During the data analysis phase, cluster searching strategies were also used to gather additional information including citation searching and author searching [40]. Such “berry picking strategies” can be helpful for knowledge building reviews because of the way they allow contextual factors to be incorporated into search strategies, particularly for reviews that are iterative in nature [28]. This kind of complementary searching can overcome the well-recognised limitations of key word and proximity searching strategies when key concepts are not clearly defined [40].

#### 2.3.2. Social Ecology Maps

During the interview study, participants were asked to complete an ecomap to describe their professional relationships in the role [41]. Participants were asked to describe relationships with the client and their family, relationships within the organisation where the case manager worked, and relationships external to where they worked. Ecomaps were used primarily to “augment” the data collected as a strategy to facilitate conversations and to gain insights into the social context that was specific to the context of each case manager [41]. The ecomaps were constructed collaboratively during the interview and the interviewer shared their screen with the participant and filled in a digital ecomap using PowerPoint while the participant was watching. The participant was also asked to identify which relationship were positive and supportive and which ones they considered stressful. A thick line indicated positive support and a dashed line was used to indicate relationships that contained tension or stress. The ecomaps were also used to assist with data analysis by helping display the number, source and quality of relationships and data organisation by grouping relationships through colour-coding to distinguish between client and family relationships, relationships within the provider organisation, interprofessional health relationships with other organisations, and intersectoral relationships with organisations outside of health. Ecomaps were anonymised.

### 2.4. Stage Four: Mapping the Data (Charting)

All included studies from health databases were first read and excerpts that were relevant to the research questions were highlighted. Studies were iteratively grouped multiple times to identify optimal ways to group the data in coherent categories. For example, the first group contained all studies that involved a nurse case manager and later iterations separated practice nurses working in a GP practice from specialist nurses linked more closely to tertiary providers (e.g., cancer nurse coordinators). A timeline was constructed to reflect the history of case management and support interpretation of the articles, in light of the clear impact on roles of changes in healthcare models and legislation. This highlighted the development of health and social navigation roles or new approaches to primary health care for example. Data from grey literature sources which included job advertisements, service information, evaluations, reports, and videos was entered into an Xcel spreadsheet. Data was charted iteratively until patterns emerged and distinctive roles and role changes over time could be identified.

Once the range of different types of case management roles in NZ had been identified then the data was synthesised for each separate role using an analytic template [28] of key concepts adapted from Lukersmith [32]. To do this one researcher (CS) listened carefully to each interview and entered a summary of the content into each analytic template. Each template also contained a list of all data sources relevant to the role. The use of an analytic template provided a way of charting specific details of each respective role while drawing on the range of different sources to support data synthesis.

A form of conceptual triangulation helped to identify convergence of ideas between sources [42]. A distinct role was identified if there were at least three different types of data available that provided convergent findings describing the role. This approach was taken to ensure conclusions between sources were consistent and could be corroborated. An overview of the different types of data sources for each role is available in Appendix A Appendix A. Preparing this overview highlighted areas where there was not yet sufficient data so two additional interviews were purposefully sought to have a more comprehensive data set (for example: interview participants 14 and 15 were interviewed to supplement the available data for the areas of mental health and older adult care management).

### 2.5. Stage Five: Collating, Summarising and Reporting the Results

The synthesised data for each distinctive role was summarised into three tables. The first table summarised the different job titles associated with each role. The second table summarised the roles and relationships that were part of the role. For the final table, we drew on theoretical concepts around continuity of care [33] and the Rainbow Model of Integrated Care [43] to identify if and how the concepts of clinical, professional, organisational and systems integration were evident in each role. More information about these tables can be found at https://cpcr.aut.ac.nz/our-research.

### 2.6. Stage Six: Stakeholder Consultation

As part of this scoping review a project advisory group was established and met at the start and again towards the end of the project. Participants were invited to be part of the group who had lived experience of working in a case management role or had significant content knowledge of the field of case management. The project advisory group included people involved in education, policy development and case management practice. Group members drew on their networks to assist with advice regarding literature searching, recruitment of participants and the significance of findings to their local context.

## 3. Results

We found a large and diverse range of sources describing and evaluating case management work in the NZ context. Figure 2 below provides an overview of the different data sources that contributed to the review. The total number of sources was 148. Within the data set there was 35 peer reviewed articles, 2 chapters, 6 theses and 28 reports and evaluations. The role that had the highest number of articles, evaluations and reports was the whānau ora navigation role, followed by the kaimanaaki role and the care manager for older adults.

The results of the review will next be described in overall terms of job titles, roles, and relationships across the data set. More information about the different types of case management, data sources, tables comparing roles, relationships, and ecology maps can be found at https://cpcr.aut.ac.nz/our-research (accessed on 9 December 2022).

This review identified at least 38 different job titles from across the data set. Few sources explained how or why the title was selected or described what it meant. Many job titles were developed in response to service innovation. For example, the Needs Assessment and Service Coordinator (NASC) role, the Whānau Ora Navigation role and more recently the adoption of a new model at ACC that led to replacing the title of ACC Case Managers with Recovery Coordinators and Recovery Partners. A further grouping of job titles used Māori language to reflect a more relational and culturally responsive role. Box 1 provides a summary of these terms accompanied by an English language explanation. Variability in job titles occurred between different roles but also within one type of role. Table 3 lists each different role type and illustrates the number of different job titles found for each role. Overall, the most frequently used job titles were those that used the word navigator or navigation in the title. The equivalent word in the Māori language is kaiārahi.

### 3.1. Job Roles

This review found sufficient data to identify 18 distinctive roles. For simplicity’s sake for this article, we have grouped these roles in 7 broadly similar areas to aid description (health and social navigation, private insurance sector, ACC, primary health care, disability and older adults, mental health, and specialist (largely clinical) roles. These groupings are based on similarities in the type of role, the context in which the case managers work and/or the type of client that they are primarily working with.

1.Health and social navigation

In this area we grouped five distinctive roles: Whānau Ora Navigators [17,44,45,46,47,48,49,50], Kaimanaaki [19,51,52,53,54,55], Māori Cancer Coordinators [56,57,58], Pacific Navigators [58,59,60] and Partnership Community Workers (PCW) [61,62,63,64]. The first four of these roles were developed in conjunction with Māori Health Providers and the fifth one was developed in Canterbury by a large primary health organisation [63]. A distinctive feature of a navigation role is the way workers support clients and their family and/or whānau (extended family) to overcome access barriers to health and social services by embracing a wide range of health and social outcomes [8,59]. Whānau Ora navigators are funded through several government departments including the Ministry of Health but delivered by Māori health providers [45]. Whānau Ora navigators focus on helping disadvantaged Māori whānau achieve their desired aspirations. Most common outcomes areas are physical health, education and/or training, whānau and/or social health [17]. Both Kaimanaaki and Māori Cancer coordinators display a similar family-centric and strengths-based ethos but focus on clients who have specific long-term conditions. The Kaimanaaki roles have been carefully developed using evidence informed practice and co-design with clients [19,52] and have demonstrated effectiveness at improving clinical and social outcomes for Māori and Pacific clients with poorly controlled diabetes who previously had limited involvement with health services [54,55]. The Māori Cancer Coordinator roles are yet to be rolled out nationwide and not mentioned in the 2019–2029 Cancer Action Plan [65]. This is despite data evaluating the role showing its value in providing continuity of care for clients, enhancing communication between health professionals and providing greater integration of care within cancer services [57]. Due to the success of Whānau Ora [45,49], new commissioning models have been developed to enable wider reach to other communities likely to benefit from the use of health navigators. Pacific Navigation roles, particularly with the provision of bilingual navigators appear to be essential to overcome cultural and systemic barriers to accessing much needed health care services for this population [59]. The fifth navigator role (PCW) was developed by a primary health organisation (Pegasus) in Christchurch to improve access to primary health care for hard-to-reach groups [61,63]. Strategic design features in the four previous roles include significant alignment between the client and family, service delivery and system level governance [19,49]. Importantly these roles are embraced by the Māori provider across all levels and have high value and visibility [46,47,49,50].

2.Private Insurance Sector

The private insurance sector commonly uses case managers to manage health insurance claims for clients who hold some form of income protection and who become disabled and unable to work. The role of the case manager is to minimise the length of time that a client is entitled to receive payouts by supporting a return to employment. In NZ there are a small number of insurance companies employing case managers [7].

3.ACC (Government provided social insurance)

Historically, ACC case managers have been the most recognised case management role in NZ [7]. This social insurance scheme provides cover for all New Zealanders in the case of injury. Recently, a system wide redesign has led to a restructure of the role on the basis of client complexity [66]. Recovery Coordinators have higher caseload numbers, but their clients are less complex and tend to have primarily physical injury claims (for example clients with disc prolapse, car crashes or treatment injuries from procedures). In contrast Recovery Partners, work with clients who are considered complex due to a serious injury (such as significant brain or spinal injuries) and require engagement with several stakeholders such as other health care practitioners (HCP) or governmental agencies. The goal of the redesign was to focus ACC resources by streamlining the system to ensure those who needed greater support could access it more easily [66].

4.Primary Health Care

The burden of managing people with often multiple long-term conditions usually falls to primary health care services and this can be a significant challenge especially given the mismatch between current funding models and client needs [67]. A common strategy is to use nurse case managers to coordinate care for those with the most complex needs [13]. These nurse case managers are often practice nurses who are unlikely to have been trained in self-management support, long term conditions or case management training and are often under significant time constraints [13].

A recent addition to the primary health team is the introduction of practitioners whose role is to provide behavioural support for health and wellbeing needs of clients in the practice [68]. These practitioners are funded through mental health budgets but geographically located in specific GP practices. The two key roles are Health Improvement Practitioners (HIP) and Health Coaches and they work as a team using a distinctive philosophy of practice [69] quite different to traditional approaches to health care. Key features include requiring a ‘warm handover’ from the GP or practice nurse, a schedule which is deliberately kept clear enough to allow for same day appointments to ensure ease of access, no follow up sessions so that future engagement is on the initiation of the client and a willingness to work with ‘what is on top’ which may include relational, health, parenting anxiety related, or lifestyle change needs. HIP roles tend to be occupational therapists with a background in mental health who mainly provide short structured sessions that provide targeted health support [70], but may liaise with the GP and secondary mental health services as needed. Health coaches provide emotional and motivational support and assistance in training in self-management and related health literacy skills [70,71]. In many ways Health Coaches function like navigators and can be very mobile in the community as they support clients to achieve their goals and navigate health and social systems. Although programme developers state these roles are not considered to be case management roles, many of the actions of health coaches can be mapped onto the case management taxonomy such as engagement, assessment, self-management skill development, navigation of health and social services and feedback to the practice team. Similarly, although HIP roles are more focused on single short intervention sessions, for some clients, coordination of mental health services will be necessary.

5.Older Adults and People with Disabilities

We found four roles in this area: Needs Assessment and Service Coordination (NASC) roles [72], Care Managers for Older Adults [16,73,74,75,76,77], Local Area Coordinator (LAC) roles [15,78] and Kaitūhuno (Connector) roles particularly for young disabled people [79,80]. The NASC role has been part of the health landscape in NZ since 1995 when the role was first introduced.

NASC roles for older adults are managed centrally through district health board funding. The purpose of the NASC role for older adults is to maximize independence and to help the person remain in their own home if possible through the assessment of needs and the provision of appropriate support. NASC services provide a centralized referral point for those who are struggling to cope in their own home by completing the assessment using the International Residential Assessment Instrument (InterRAI). This tool also provides specific instruction and criteria for determining the amount of support an older adult is eligible to receive. Examples of DHB funded home and community support includes personal cares, equipment, household support, carer support, respite care services and therapy assessment services [72]. NASC services have been criticized for focusing on narrow and fixed western perspectives of support [81] and there is a lack of transparency for clients regarding the assessment process. We found limited literature describing or critiquing this role [72]. Officially NASC roles do not provide a consistent named person and relational continuity. However, interviews with two NASC workers in this study indicated that at times informal relational continuity did occur with negotiation. Additionally, the structured assessment processes did provide informational and management continuity, so we elected to retain this role in the study concluding that NASC roles provide a type of light touch case management that largely uses a brokerage model with limited relational continuity.

In NZ, an enablement model known as a restorative home support (RHS) model, has been developed to enhance the capability of older adults to live in their own home for as long as possible [76]. Key features of this model include: a care manager enhancing professional integration by being in regular contact with the client’s GP, careful coordination of services, targeted holistic assessments, structured goal setting with small achievable steps and regular review; and home-based physical training with assistance [76]. In this review, we found evidence that interventions that used these elements produced very positive outcomes including an increase in the length of time older adults could remain in their own home and extended survival rates [16,74,75]. Despite these promising results, we found limited evidence in our review that these results had been implemented nationwide into routine practice with few people employed as care managers for older adults. Instead, we found some evidence of significant unmet needs and deep distress from older adults and their families, in need of assistance with navigating, negotiating, and coordinating care for themselves [82].

In relation to services for people with disability, we identified two related navigation roles The first is called a ‘Local Area Coordinator’(LAC) whose role is to support people with disabilities to become more independent and to achieve their desired goals by connecting them with community resources that can provide additional support [15,78]. Such community connections can include different industries, education, or other community-based resources. In this role, the LAC’s credibility comes not from professional qualifications but from the quality and quantity of community connections and their ability to provide motivational and emotional support [15]. The LAC role was developed as a pilot in the Bay of Plenty and based on an Australian model. Nationally, there has been debate over how to better provide choice and control for people with disabilities, and the Enabling Good Lives principles (EGL) are increasingly recognised as providing standards for best practice [79,80]. These principles are underpinned by the importance of self-determination, the need to support disabled people to work toward their aspirations and goals, and the recognition that support for people with disabilities needs to be individually tailored, flexible and strengths-based. A new navigator role called a kaitūhuno (connector) is being developed. A distinctive feature of the kaitūhono (connector) is the prioritisation of lived experience over clinical or community connections as a way of providing both credibility and enhancing engagement with clients [79,80].

6.Mental Health Practitioners

Our scoping review found limited literature about case management for clients with mental health conditions in NZ, so most information was gained through two interviews and job advertisements. There were two main categories of mental health practitioners doing case management work in this area: registered health professionals (usually social work or occupational therapists) and community support workers. Both groups were involved in planning and organising care, but community support workers had a smaller percentage of their role devoted to care coordination, and a higher proportion devoted to direct delivery of care. This contrasted with registered health professionals who had more complex clients and did more care coordination. For both groups a significant amount of emotional and motivational support was provided to clients using a strength-based and assertive model of practice.

7.Specialist roles

The final areas include two roles that require specialist clinical skills. The Cancer Nurse Coordinator (CNC) role was developed as a nationwide attempt to improve care coordination, psychosocial support, and information support for clients with cancer [83,84,85]. One nurse with advanced nursing skills was appointed to the role for each district health board. An evaluation of the role showed it was very effective at improving access and timeliness to treatment services, delivering high levels of client satisfaction, and the CNC also helped to identify areas of improvement along the cancer continuum [85]. One limitation of the role was that Māori and Pacific clients with cancer had lower levels of uptake than anticipated. The Cancer Action Plan (2019–2029) does not include any information about this role [65].

In NZ independent midwives can become a lead maternity caregiver (LMC) and are the first choice for the majority of NZ women who are pregnant [86]. Midwives have a strong emancipatory philosophy and use an explicit partnership model that includes advocating for the mother, baby, and extended family [87]. A case loading model is the usual approach to care and some midwives use aspects of a case management process with clients with whom they feel would benefit from additional advocacy. Examples of case management actions for the two Māori midwives in our study included high levels of engagement, ongoing assessment of wellbeing, emotional, and motivational support, training in relevant mothering skills like breastfeeding, education, advocating and coordination of pregnancy-related care [4]. We found limited information in the midwifery literature that explores these case-management related actions that are part of using a case management approach [86,87]. The role is very mobile and can be particularly intensive especially if complex social issues emerge. The Māori midwives in our study appear to use a model that combines clinical, strength-based, and intensive approaches to care.

### 3.2. Relationships

The consistent finding from the ecology maps and literature was the number and variety of relationships that case managers needed to manage. Some roles had comparatively less relationships (such as case managers for private insurance) and others (who had more complex clients) had greater number of connections. Relationships could be divided into interprofessional relationships within the provider organisation, relationships with other professionals from different organisations, and intersectoral relationships with a range of social services that included governmental and non-governmental organisation.

The Rainbow Model of Integrated Care proposes that integration needs to be considered across multiple levels; at the level of the relationship with the client and their family (clinical integration); between different health care practitioners (professional integration); between various organisations (organisational integration); and ideally at a systemic level with the alignment of rules and policies [43,73]. Using this model helped to make sense of the different types of relationships. An example of the ecology map of an ACC Recovery partner is shown below in Figure 3. Each relationship is colour-coded to reflect the different types of relationships: blue for relationships with client and family; orange for relationships within the organisation; green for relationships with health care practitioners in other organisations and grey for intersectoral relationships with organisations in the social sector. Strong supportive relationships have a bolder outline and relationships which have a tenuous or stressful element have a dashed line. The distance from the central bubble does not indicate any relative importance or size, merely how each relationship could be fitted in to a two-dimensional graph.

As well as relationships with clients and families, case managers also had multiple relationships with health care practitioners across different organisations (GP practices, specialist medical providers, DHB services (inpatient and outpatient), home-based support providers and others). Relationships from the social sector including governmental organisations like ministries of children, housing, income, social development, and several non-governmental organisations that provided relevant services. A further distinctive feature was differing sources of tension and sources of support in relationships. For example, the case manager in private insurance indicated stressful relationships with clients, GP’s, and specialists. Given the inherent tension in navigating funding, eligibility and return to work this is unsurprising. For another case manager who was navigating health and social factors; a limited amount of suitable housing options produced stressful relationships with intersectoral providers of housing. For the recovery partner, working with specialists who were not aligned with or contracted to ACC was difficult. Taken as a whole, these maps reflect the important relational work done by case managers to ensure continuity and coordination of care.

## 4. Discussion

This scoping review uncovered a substantial body of NZ case management literature The 38 different job titles highlighted even greater terminological variance in NZ than found in international reviews. For example, one study found 11 different job titles in an Australian setting [20] and another listed 29 different terms to describe health navigation roles [59]. Two of the most common terms, often used interchangeably in the international literature, are ‘case management’ and ‘health navigation’. One comprehensive scoping review carefully compared the training, knowledge, actions, and clients of case managers and health navigators and concluded that, though there was considerable overlap in many aspects, the only distinctions between the two roles was that case managers tended to have a more clinical focus to their work and often had professional training such as nursing or social work [88]. In our NZ study, only about a third of the roles had a clinical focus and the majority aligned more closely with health navigation. Over time, fewer NZ organisations have retained the job title ‘case manager’ including ACC, who has recently adopted more client-centered terms. The title ‘navigator’ was also the preferred term of the project advisory group over ‘case manager’ because it more closely aligned with their experiences working in the field. One simple strategy to help demystify case management in NZ would be to shift toward using more consistent terminology for job titles. Consistently adopting the title ‘navigator’ (or kaiārahi in Māori) could reduce some of the role confusion and the variation in job titles in the NZ setting.

A further key finding of this study was the degree of innovation and adaptation of the role. This did make it challenging for the study team to identify what roles could be described as case managers and what did not. As a ‘rule of thumb’ we chose to include a role if it had aspects of all the following four elements: (1) there was a named person (or team); (2) the case manager provided some degree of relational, informational or management continuity of care; (3) the case manager did some planning and organising of care coordination. and (4) the case manager did several (but not necessarily all) of the actions listed in the Lukersmith case management taxonomy [4]. This meant that roles that had no continuity of care and focused purely on coordination or only involved direct delivery of care (such as many support worker roles) were not included. Innovative adaptations were evident across all elements. Some approaches provided continuity of care through paired teams such as the use of a nurse case manager and cultural support worker (kaitautoko) [18]; and health improvement practitioner and health coaches [68]. Other roles aimed to explicitly minimise or reduce relational continuity (e.g., NASC roles). Some roles delivered care as well as coordinated some aspects of care (such as community support workers in mental health). Some roles divided the co-ordination work based on complexity such as ACC recovery coordinators and recovery partners [66]. Although seldom explicitly acknowledged, the underlying driver of service adaptation is likely the implicit recognition that case management work is too demanding for one individual given the time, energy, and relational skills needed. Innovative adaptations may be the result then of the ongoing tension between service quality and affordability inherent in case management work. This review tries to make more visible key shared components that underpin case management work and highlight innovative aspects of service delivery that provide points of difference to make this area easier to understand for practitioners, educators and policy makers.

One role, the whānau ora navigator, stood out for both the quantity and quality of data describing the role. The literature described the development, underlying theories of change, and transforming impact on clients and their families/whānau [17,44,45,47,49,50,81,89]. Of peer reviewed articles and government publications, 18/71 (25%) related to Whānau ora navigators (or kaiārahi) and described how their activities make pivotal contributions to an innovative and highly effective programme of health care that is well-recognised internationally as a public health success story [49]. Whānau ora navigators are described as the backbone of whānau ora initiatives [17] and are highly visible and valued in Māori health organisations who employ them [50,55]. A theory of change explains how an intervention or programme works to achieve desired impact [12]. Whānau ora navigators use a relational process to achieve gains [44] which is described Figure 4. The model combines strength-based and intensive approaches, culturally appropriate engagement, whānau led and whānau driven aspirational goal setting. Key cultural elements are reflected in the way services are delivered, including wrap-around support (in contrast to narrow western concepts of support), mutual reinforcement of shared values (including exploring shared social, genealogical and regional connections), a holistic wellbeing focus for all the family, and active building of connections between whānau members with an expectation of reciprocity [44,47,81,90]. The way that Māori have embraced this relational navigation role and the value it holds to them provides a contrast to the lack of visibility and value for many of the other roles in this review. More information about this role and how it was developed is summarised in this case study by the Productivity Commission report [45].

Another striking finding of this study is the diverse and extensive inter-professional, inter-organizational and inter-sectoral relationships reported by interview participants and supported by qualitative data that mentioned different types of relationships [13,44,51,57,59,60,61,63,81]. This research is one of the first to consider the relational contribution case managers make and the findings are marked, reinforcing the earlier contention that case managers seem to provide a type of relational glue that is necessary for integrated care [1]. For example, prior research has focused primarily on the actions of case management work, and aside from client-related actions to support engagement, there is minimal descriptions of the types of social networks needed in the role [4]. The use of socio-ecology maps has a long history in qualitative research [41]. We found significant methodological challenges though describing the number, type and quality of relationships and trying to report these social networks using the ecomap as a sole data source. Case managers due to their extensive social networks appear to be the quintessential interprofessional practitioner. Their ecomaps also reported differing sources of interpersonal stress depending on their respective roles. The participant working in private insurance described tension in their role enforcing the contractual entitlements of the business and the needs of clients. The participant working as a practice nurse case manager described strong and supportive relationships with community pharmacists but stressful relationships with hospital needs assessors. (see https://cpcr.aut.ac.nz/our-research for more ecomap exemplars- accessed on 9 December 2022) These findings indicate the need to consider a broader range of relationships beyond those with other health care professionals and to also explore consider new ways of examining the significant relational contribution and tensions inherent in this role.

### 4.1. Limitations

There are some significant limitations to this scoping review. Although we aimed to provide a comprehensive overview it is likely that some roles or job titles were omitted. We did gather some data on two roles which did not have sufficient information to meet our threshold criteria to be included (ACC Navigators, and COVID related public health roles). This study focused solely on health-related roles which meant similar roles in the social sectors such as vocational case managers or correctional case managers are missing from this study. The study design does not enable us to estimate the number of people working in this area in NZ. This study is an initial exploratory review and is the first of its kind in the NZ context. We encourage others to continue to explore this field and notify the authors if additional roles emerge in this field over time so that we can continue to build an accurate picture of this kind of work in the NZ setting.

### 4.2. Implications

There are several implications from this review for policy makers, educators, and practitioners. Changes to public policy need to be made to improve recognition and visibility of this role. Around 40% of the current health care workforce are non-regulated [3] because they are not covered by the HPCA Act [91]. There is only limited workforce data about this sizeable section of our health workforce [2]. With regard to the information about case managers, care coordinators and health navigators- even less is known. Health Workforce NZ themselves have said that case managers “are not considered to be a distinct occupation or qualification grouping” as an explanation for why they are not listed in their workforce documentation (P. Maciver Personal communication 13 October 2021). The Kaimanaaki Workforce Plan [27] focuses only on the social sector and neglects case managers working in health. A core finding of the NZ Productivity Commission [21] was the need for a navigator to help vulnerable clients to access health and social services. The first step in workforce development must be the explicit inclusion of health navigation roles in public policy and related workforce development plans. Having a consistent term such as health navigator could help reduce role confusion. Secondly, this review has highlighted that despite the variation in contexts, there are several components of case management work that are shared [34]. The next step for those involved in education is to work toward the development of explicit competency frameworks to make visible the core and transferable skills required to effectively work in this complex area and provide training to support workforce recruitment, retention, and professional development. Finally, our desire is that this review will add value to several case management practitioners in this country working tirelessly by highlighting their unique contribution to the health and wellbeing of so many.

## 5. Conclusions

A rich and diverse body of literature describing and evaluating 18 distinctive case management roles in New Zealand was uncovered including 35 peer reviewed articles, 2 chapters, 6 theses and 28 evaluations and reports. Social ecology maps highlighted diverse interprofessional and intersectoral relationships. Whānau Ora navigator roles were highly valued, embraced by Māori and considered to the be backbone of an international public health success story. In contrast most other case management roles lacked role clarity and visibility in the NZ health system. Nationwide, significant innovation and adaptation is evident in this field; particularly in the last four years. Case managers in NZ play a pivotal, but largely unappreciated role and their work is often unrecognised. This seriously impedes workforce planning and optimisation of integrated care strategies.

## Figures and Tables

**Figure 1 ijerph-20-00784-f001:**
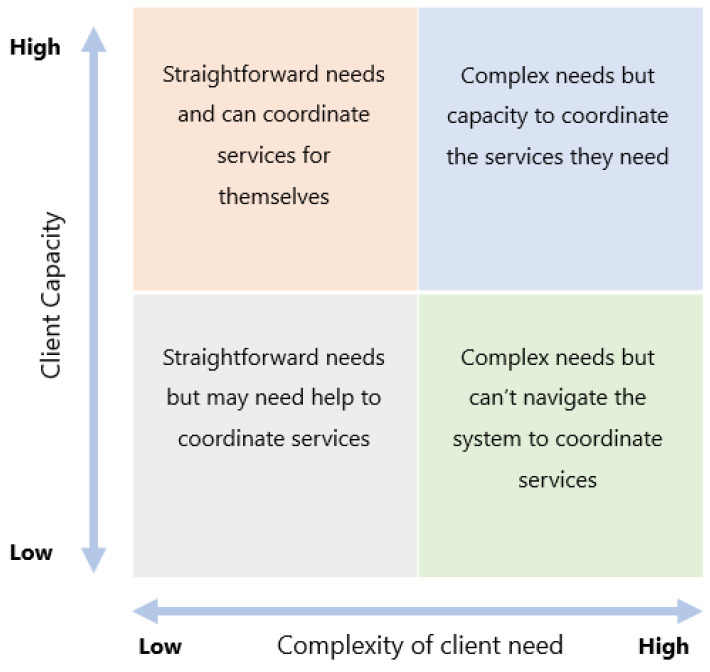
Matrix of complexity, client needs and capacity. (Permission to reproduce granted by the NZ Productivity Commission).

**Figure 2 ijerph-20-00784-f002:**
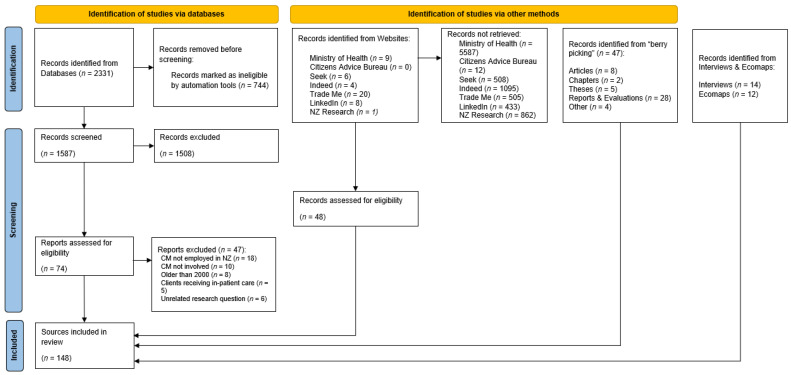
Flow of information through the review: PRISMA.

**Figure 3 ijerph-20-00784-f003:**
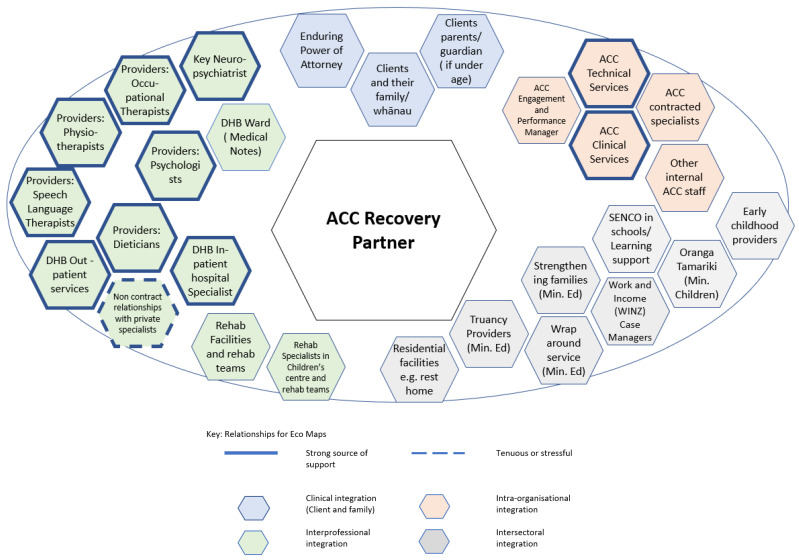
Example of ecomap for ACC Recovery Partner who focuses on child and adolescent clients.

**Figure 4 ijerph-20-00784-f004:**
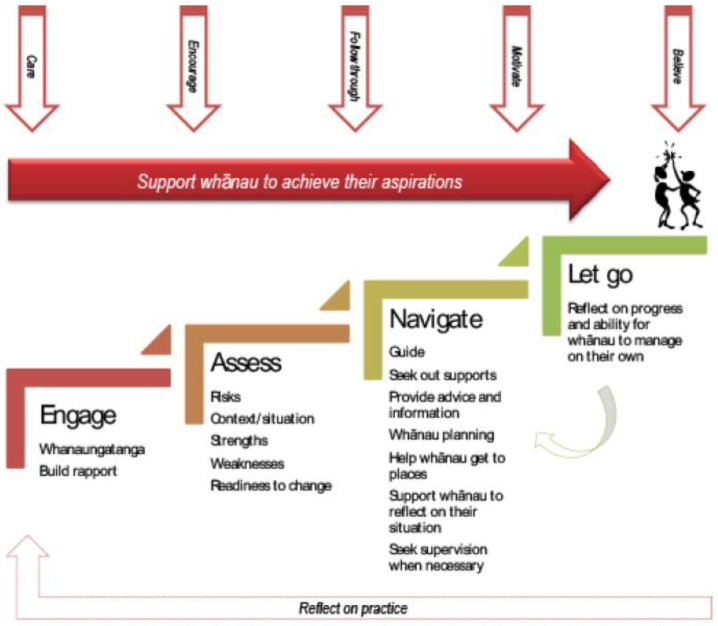
The navigational approach of Whānau ora navigators (reproduced with permission); taken from [44].

**Table 1 ijerph-20-00784-t001:** Concepts underpinning case management roles.

Continuity of care [33]	Relational continuity Informational continuity Management continuity
Role Purpose [4,32]	Client and whānau (extended family) relationships Client circumstances and characteristics Theoretical frameworks influencing practice Interprofessional and intersectoral relationships Service context and jurisdiction Mobility/intensity of role
Role function/actions (from taxonomy) [4]	Engagement Emotional and motivational support Education Advising Monitoring Holistic assessment Planning Coordination (including navigating, facilitating, advocating, collaboration, case consultation, managing documentation)

**Table 2 ijerph-20-00784-t002:** Overview of Interview Participant’s Characteristics.

Participant	Area	Organisation Type
1	Pacific Navigation	Pacific Organisation
2	Other—Case Load Midwife	Māori Organisation
3	Other—Case Load Midwife	Māori Organisation
4	Accident Compensation Corporation (ACC) Recovery Coordinator	Governmental Social Insurance Provider (ACC)
5	Behavioural Support	Primary Health
6	Needs Assessment Service Coordinator (NASC)	District Health Board (DHB)-(Waitemata Older Adults)
7	Practice Nurse	Primary Health
8	Mental Health	Māori Organisation
9	Behavioural Support	Primary Health
10	NASC	DHB (Counties- Disabled and Older Adults)
11	Private Insurance	Private Company
12	Other: Case & Contact Tracer	Public Health
13	ACC—Recovery Partner	Governmental Social Insurance Provider (ACC)
14	Mental Health	Non-Governmental Organisation (NG0)
15	Other: Specialist Older Adults	NGO

**Table 3 ijerph-20-00784-t003:** Summary of Roles and Job Titles.

No.	Name of Role (*n* = 18)	Context	Other Job Titles (*n* = 38)
**1**	Whānau Ora Navigator	Health & Social Navigation	Kaiārahi; Paeārahi; Tiwhana Navigator; Kaiwhakaaraara; Kaiwhakahere; Kaimahi
**2**	Kaimanaaki	Health & Social Navigation- Diabetes/Long Term Conditions	Kai Manaaki; Kaimanaakiwhānau; Kaitautoko
**3**	Māori Cancer Coordinator	Health and Social Navigation-Cancer	Kaimanaaki; Mate Pukupuku; Māori Cancer Kaiārahi
**4**	Pacific Navigator	Health & Social Navigation-culture specific	Wellbeing Community Connector; Whānau Ora Navigator; (Pacific); Navigator
**5**	Partnership Community Worker (PCW)	Health & Social Navigation	
**6**	Case Manager (Private Insurance)	Private Insurance	
**7**	ACC Recovery Coordinator	Social Insurance—ACC	Case Manager
**8**	ACC Recovery Partner	Social Insurance—ACC	Case Manager
**9**	Practice Nurse Case Manager	Primary Health Care	Community Nurse; Outreach Nurse
**10**	Behavioural Support: Health Improvement Practitioner (HIP)	Primary Health Care	
**11**	Behavioural Support: Health Coach	Primary Health Care	
**12**	Needs Assessment Service Coordinator (NASC)	Disability sector/Older Adults	
**13**	Local Area Coordinator (LAC)	Disability	
**14**	Kaitūhuno (Connector)	Disability	Independent Facilitator; Local Area Coordinator; Navigator; Community Connector
**15**	Care Manager Older Adults	Older Adults	Case Manager; Nurse Care Manager; Health Professional Care Manager; Care Coordinator; Homecare Coordinator; Clinical Manager
**16**	Cancer Nurse Coordinator (CNC)	Other/Specialist: Cancer Nursing	Cancer Nurse Specialist; Nurse Case Manager- Long Term Conditions (Cancer)
**17**	Case Loading Midwives	Other/Specialist: Community Midwives	Māori Midwife; Lead Maternity Caregiver (LMC)
**18**	Key Worker: Mental Health	Other/Specialist: Mental Health	Case Manager; Navigator; Community Support Worker; Kaiwhakahaere; Registered Health Professional; Pathways Navigator; Mental Wellbeing Navigator

## Data Availability

Additional information about the findings of this study and the data set can be found at https://cpcr.aut.ac.nz/our-research (accessed on 9 December 2022).

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
