# Peer review of "Demystifying Case Management in Aotearoa New Zealand: A Scoping and Mapping Review"

_ijerph, 2022, doi:10.3390/ijerph20010784_

Round 1

Reviewer 1 Report

This scoping review aims to map with landscape of case management work to clarify job roles and responsibilities associated with the the terminology of 'case management'. The topic of the paper is quite niche but has some relevant implications and is well structured and the overall methodology is well-described. Please find my comments and suggestions for minor revisions below:

I suggest to re-write the Abstract in structured form. I suggest to include headings: Background, Aim, Methods, Results, Limitations, Conclusion. 

line 42: case management role in NZ appears to have evolved differently - it would be nice if you were able to describe why this role might have evolved differently to other countries and what the implications are.

Lots of case management happens in the private health sector, such as lived-experience organisations who often offer case management - how do you think does the exclusion of these in your search influence your observations? Maybe worth a discussion point. 

Figure3: I recommend to include the colour-coding in a legend for the figure and the information what line thickness represents (as explained in lines 255/256) so that the figure can be self-standing and self-explanatory.

Do the relationships represented in the ecomaps reflect subjective experience of relationships (i.e., individuals have personal relationships with other service providers) or are these objective observations by the interviewee? I could imagine that personal relationships play a huge role in the successful delivery of support. Maybe worth to add this to the discussion.

The overall implications of a lack of a coherent terminology could be carved out in more detail, in both the Introduction as well as the Discussion. What are the exact implications of your findings? The message of the manuscript could be strengthened by moving the 'Implications' section into the focus of the discussion and making the need for a consistent terminology clearer in the introduction.

While not necessarily within the scope of this review, what is the significance of case management for Mãori culture and non-Indigenous NZ individuals? 

Author Response

Thank you for your helpful comments. Please see the attached file. Our responses are also recorded below

Comment

Comments/Thoughts/Responses

Revisions

I suggest to re-write the Abstract in structured form. I suggest including headings: Background, Aim, Methods, Results, Limitations, Conclusion. 

Thankyou. We have now included headings to improve the structure

line 42: case management role in NZ appears to have evolved differently - it would be nice if you were able to describe why this role might have evolved differently to other countries and what the implications are.

The main difference in NZ according to Currie 2012 is that case managers are less well trained essentially and don’t necessarily have a background or qualification in health, and there are fewer educational options available.

Thank you. We have added in the following to try to address this comment:

In contrast in NZ, having a recognised health qualification or experience is not a requirement for employment and fewer options for workforce training are available [7]. 

Lots of case management happens in the private health sector, such as lived-experience organisations who often offer case management - how do you think does the exclusion of these in your search influence your observations? Maybe worth a discussion point

I wonder if you are referring to NGO type providers, trusts for example as the private health sector? NGO providers are more likely to include or attract some people with lived experience. Our search was as broad as possible so should have picked up anyone doing health navigation /connector roles. The kaituhuno /connector role for example is often someone with a lived experience. In NZ most health funding comes via District Health Board ( Ministry of Health funding) for the most part so there is not a lot of funding available for lived experience roles in our country I think

Figure3: I recommend to include the colour-coding in a legend for the figure and the information what line thickness represents (as explained in lines 255/256) so that the figure can be self-standing and self-explanatory

Great feedback

Thank you. Key for Figure 3 is now included

Do the relationships represented in the ecomaps reflect subjective experience of relationships (i.e., individuals have personal relationships with other service providers) or are these objective observations by the interviewee? I could imagine that personal relationships play a huge role in the successful delivery of support. Maybe worth to add this to the discussion.

They are definitely subjective experiences although given the constraints and tensions and expectations of each different role there is a consistent logic to the experience ( the tension is clearly explainable and related to the role expectations and not related to individual characteristics of the person)

I have included into the discussion some additional information and exemplars to illustrate this

Their ecomaps also reported differing sources of interpersonal stress depending on their respective roles. The participant working in private insurance described tension in their role enforcing the contractual entitlements of the business and the needs of clients. The participant working as a practice nurse case manager described strong and supportive relationships with community pharmacists but stressful relationships with hospital needs assessors. ( see https://cpcr.aut.ac.nz/our-research for more ecomap exemplars)

The overall implications of a lack of a coherent terminology could be carved out in more detail, in both the Introduction as well as the Discussion. What are the exact implications of your findings? The message of the manuscript could be strengthened by moving the 'Implications' section into the focus of the discussion and making the need for a consistent terminology clearer in the introduction

You make a good point that making a stronger case for more coherent terminology is warranted

Added into introduction and implications section  more explicit comments about consistent terminology to make this message a little clearer ( see highlighted in text)

While not necessarily within the scope of this review, what is the significance of case management for Mãori culture and non-Indigenous NZ individuals? 

The short answer is that we are still building knowledge in this area. This article brings together a surprising number of sources that reflect the way that indigenous models of case management are highly valued and have strong evidence.  This article should enhance the profile of this kind of work which is not as widely recognised as it deserves to be. It also highlights the relational contribution that underpins Maori approaches. For non-Maori with complex health needs a relational approach is also important but often this is  not acknowledge in policy or planning. As is often the way what is good for Maori health is good for all, and this article introduces the relational contribution that underpins all different case management roles

Reviewer 2 Report

Dear authors,

Thank you for the opportunity to review your interesting manuscript. I will give my feedback following the structure of the manuscript. 

1.Title and abstract

The title is informative and the abstract provides a summary of the manuscript's major aspects.

2.Background

I would like to congratulate the authors for this chapter. From my point of view it contains valuable information and is very well written and well organized.  However,it might be too  long, therefore the authors could see if it needs to be shortened a bit.  

Another consideration is to explain what Maori is. I understand that’s the native language of  New Zealand. However it might be positive to explain it. 

3.Material and Methods

The chapter is very clear and well-written. I only have some suggestions for the authors. 

-It important to clarify the meaning of the PCC the first time the concept appears (line 144: “The PCC framework…”)

-In table 1 there is information on the references used in both the topics continuity of care and role function. However there isn’t information from the reference about the purpose of the role. It’s right?

-Stage two, Relevant sources → Here the authors explain that they interviewed case management nurses, but we have no information about how these case management nurses were recruited, how many case management nurses participated, as well as some sociodemographic and occupational information about them. From my point of view, this information is necessary.  In addition, I don’t know if the information from lines 175 to 177 it’s necessary “The student (WYC) was supervised by a multidisciplinary team of two academics with interest in this field (CS, DW) and a community associate who provided us with cultural support. A review protocol for the project was written but not published.”

-Stage three → I have the same problem here. Is it necessary to point out the role of the authors through the different sections? In my opinion this information should appear at the end of the article, in a specific section about this information (ex. line 194: WYC, DW;  line 191: WYC). 

-Interview data→ I found here that information missed in stage two section. Couldn’t the information from these two sections  be combined?

-Stage four. The same suggestion regarding the author contribution (line 280:CS).

4.Results

This chapter is very clear, very well-written and very interesting. So I would like to congratulate the authors for their work. I only have a suggestion about figure 3. Although the authors explain the different meanings in each figure and color. Wouldn’t it be nice to add a little legend? 

5.Discussion and Conclusions

No suggestions for the authors. From my point of view both chapters are very clear and well-written. 

Finally, I would like to congratulate the authors for this work.

Author Response

Thank you for your comments. Please see our response below.

Review comment

Our comment

1.Title and abstract

The title is informative and the abstract provides a summary of the manuscript's major aspect

Thank you

I would like to congratulate the authors for this chapter. From my point of view it contains valuable information and is very well written and well organized.  However, it might be too  long, therefore the authors could see if it needs to be shortened a bit.  

Another consideration is to explain what Maori is. I understand that’s the native language of  New Zealand. However it might be positive to explain it. 

I have added this in to make it clear what this word refers to.

Māori are the indigenous people of New Zealand.

- The chapter is very clear and well-written. I only have some suggestions for the authors It important to clarify the meaning of the PCC the first time the concept appears (line 144: “The PCC framework…”

I have reordered the sentence to make it clearer

The PCC framework was used to identify the population (P), concepts (C ) and context (C ) of this study [31] and help develop the  overall research question and inclusion criteria.

-In table 1 there is information on the references used in both the topics continuity of care and role function. However there isn’t information from the reference about the purpose of the role. It’s right?

Thankyou – I have rechecked and added in appropriate references

Role Purpose [4,32]

-Stage two, Relevant sources → Here the authors explain that they interviewed case management nurses, but we have no information about how these case management nurses were recruited, how many case management nurses participated, as well as some sociodemographic and occupational information about them. From my point of view, this information is necessary.  In addition, I don’t know if the information from lines 175 to 177 it’s necessary “The student (WYC) was supervised by a multidisciplinary team of two academics with interest in this field (CS, DW) and a community associate who provided us with cultural support. A review protocol for the project was written but not published.”

In Table 2 we provided an overview of the 15 interview participants, the area they worked in and the organisation type. Participant number 7 was a practice or case management nurse so there was just one nurse. In line 228 we note that participants were recruited through personal and professional networks. Participant 7 was part of our advisory group. Unfortunately, we did not routine collect sociodemographic information about them which was possibly an oversight

Content shifted earlier in the draft as suggested by the reviewer later in this document.

Deleted and put into the author contribution section in the end notes

-Stage three → I have the same problem here. Is it necessary to point out the role of the authors through the different sections? In my opinion this information should appear at the end of the article, in a specific section about this information (ex. line 194: WYC, DW;  line 191: WYC). 

Agreed; removed

;

-Interview data→ I found here that information missed in stage two section. Couldn’t the information from these two sections  be combined?

Agreed; this information has been moved earlier

-Stage four. The same suggestion regarding the author contribution (line 280:CS).

Removed. Thank you

Results: This chapter is very clear, very well-written and very interesting. So I would like to congratulate the authors for their work. I only have a suggestion about figure 3. Although the authors explain the different meanings in each figure and color. Wouldn’t it be nice to add a little legend?

Thank you for your positive comments;

Absolutely, the other reviewers also suggested this. A legend has been added

Key for Figure 3 is now included

5.Discussion and Conclusions

No suggestions for the authors. From my point of view both chapters are very clear and well-written.

Finally, I would like to congratulate the authors for this work.

Thank you!